# Sinkhorn doubly stochastic attention rank decay analysis

## Abstract

The self-attention mechanism is central to the success of Transformer architectures. However, standard row-stochastic attention has been shown to suffer from significant signal degradation across layers. In particular, it can induce rank collapse, resulting in increasingly uniform token representations, as well as entropy collapse, characterized by highly concentrated attention distributions. Recent work has highlighted the benefits of doubly stochastic attention as a form of entropy regularization, promoting a more balanced attention distribution and leading to improved empirical performance. In this paper, we study rank collapse across network depth and show that doubly stochastic attention matrices normalized with Sinkhorn algorithm preserve rank more effectively than standard softmax row-stochastic ones. As previously shown for softmax, skip connections are crucial to mitigate rank collapse. We empirically validate this phenomenon on both sentiment analysis and image classification tasks. Moreover, we derive a theoretical bound for the pure self-attention rank decay when using Sinkhorn normalization and find that rank decays to one doubly exponentially with depth, a phenomenon that has already been shown for softmax.

## 1 Introduction

Transformers are the state-of-the-art architecture for large language models and have proven successful across a wide range of domains, going from natural language processing (Tunstall et al., 2022) to computer vision (Dosovitskiy et al., 2020). Since their introduction in (Vaswani et al., 2017), the self-attention mechanism, originally inspired by (Bahdanau et al., 2014), has been the subject of extensive theoretical and empirical investigation. In standard implementations, the attention matrix is row-stochastic: each row sums to one and can therefore be interpreted as a discrete probability distribution describing how strongly a given token attends to all other tokens in the sequence. This enables the model to learn pairwise token interactions, with the objective of extracting meaningful representations that capture the underlying structure of the input data (Khan et al., 2022).

Standard row-stochastic self-attention exhibits two major shortcomings in signal propagation across the layers of the Transformer architecture (Giorlandino & Goldt, 2026). The first shortcoming concerns the rank collapse as depth increases. Indeed, Wang et al. (2020) observe that self-attention matrices are inherently low rank, with the effective rank decreasing as layer depth increases. Building on this observation, Dong et al. (2021) show that in a pure self-attention network, with feed-forward layers and skip connections disabled, the entire network output converges to a rank-one matrix doubly exponentially with depth. This rank collapse progressively destroys the input sequence information, by reducing it to uniform token representations. Notably, skip connections are shown to play a key role in mitigating rank collapse, as further observed in (Noci et al., 2022; Wang et al., 2022). The second shortcoming is related to entropy collapse. Zhai et al. (2023) demonstrate that the Shannon entropy of the attention distribution decreases during training, resulting in highly concentrated attention scores in which each query attends to only a small subset of tokens. This entropy collapse results in suboptimal information flow and, more importantly, leads to high training instability. A common strategy to mitigate this issue is to increase the softmax temperature, which smooths the attention distribution and increases its entropy (Agarwala et al., 2020; Xuan et al., 2025).

When examining the evolution of self-attention during training, Sander et al. (2022) notice that the attention matrix tends to converge to a doubly stochastic matrix as the number of epochs increases, despite

being row-normalized via softmax by default. Motivated by this behaviour, they constrain the attention matrix to be doubly stochastic since the beginning of the training, by replacing softmax with Sinkhorn algorithm (Sinkhorn, 1964; Sinkhorn & Knopp, 1967). The resulting architecture is referred to as Sinkformer. This can improve the performance. Following (Sander et al., 2022), a growing body of work has replaced vanilla row-stochastic self-attention with doubly stochastic normalization, consistently reporting its performance advantage, see for instance (Kim et al., 2024; Shahbazi et al., 2025; 2026; Born et al., 2025). Intuitively, doubly stochastic attention has a similar effect to increasing the entropy of the attention distribution, so that less interactions are missed and tokens attend more uniformly to one another (Born et al., 2025), see Figure 1.

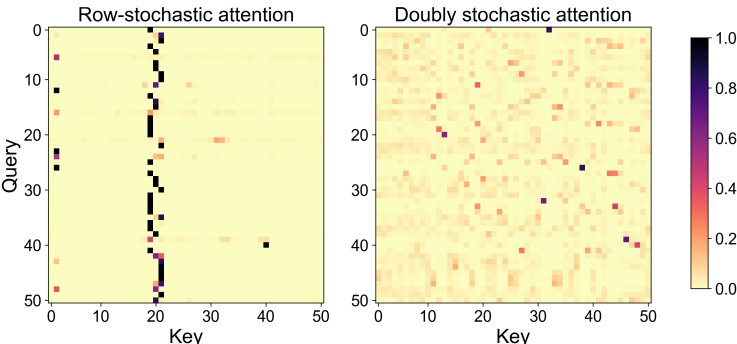

Figure 1: Attention matrices from the first layer and a single attention head of a Vision Transformer Dosovitskiy et al. (2020) trained on MNIST Deng (2012), for one sampled input image. See Appendix F for details on the experimental setup. Row-stochastic attention (softmax) concentrates on a few key tokens, while doubly stochastic attention (Sinkhorn) distributes attention more uniformly across tokens. Both matrices are visualized on a shared color scale for direct comparison.

In (Sander et al., 2022), a connection between self-attention and optimal transport is also established. The Transformer is viewed as a model acting on discrete probability distributions and the infinite depth limit of a sequence of attention blocks is analyzed, the output given by solving a neural ODE (Chen et al., 2018). Under a symmetry assumption for the query and key matrices, Sinkhorn normalization enables the iteration of self-attention layers with skip connections to be interpreted as a Wasserstein gradient flow for an energy minimization, while this does not apply to softmax. Indeed, doubly stochastic attention can be viewed as defining a transport plan between queries and keys, see (Shahbazi et al., 2025), with Sinkhorn algorithm solving the entropy-regularized optimal transport problem with an improved complexity (Cuturi, 2013) compared to a linear program (Gabriel & Marco, 2019).

Despite being a natural choice, several works have pointed out limitations of using Sinkhorn algorithm to enforce doubly stochasticity in self-attention. First, its iterative nature can be computationally expensive (Shahbazi et al., 2025), and the optimal number of normalization iterations is typically determined empirically rather than theoretically (Born et al., 2025). Additionally, poor initialization can significantly deteriorate performance (Thornton & Cuturi, 2023). To address these issues, alternative approaches have been proposed. Focusing on computational efficiency, Shahbazi et al. (2025) replace iterative Sinkhorn normalization with a mechanism based on sliced optimal transport (Rabin et al., 2011; Kolouri et al., 2015; 2019), specifically leveraging expected sliced transport plans as introduced in (Liu et al., 2024). Moreover, Born et al. (2025) exploit a direct link between doubly stochastic matrices and unitary operators to propose an hybrid quantum-classical doubly stochastic Transformer, which replaces the non-parametric Sinkhorn algorithm with the variational quantum circuit in (Mariella et al., 2024).

In this paper, we study the rank collapse of self-attention along the Transformer depth, and we compare the row-stochastic with the doubly stochastic case. To the best of our knowledge, this is the first study of this comparison. Since our focus is not on empirical performance but on structural properties, and given its widespread use in practice, we enforce doubly stochasticity using Sinkhorn algorithm. Both in our theoretical analysis and empirical evaluation, we build on the path decomposition framework of (Dong et al.,

2021), where a multi-head self-attention network is decomposed into a linear combination of *paths*, each path corresponding to a sequence of single-heads across layers, one head per layer (see Section 2 for more details). Our main contributions are as follows:

1. We derive a norm bound on the distance between the output of a pure self-attention network and the limit rank-one matrix of uniform representations, when enforcing doubly stochasticity with Sinkhorn algorithm. As shown for softmax in (Dong et al., 2021), rank collapses doubly exponentially with depth.

2. We experimentally measure rank decay in a Transformer architecture trained from scratch and show that enforcing doubly stochasticity significantly delays the collapse compared to the row-stochastic case, see Figure 2. We note that skip connections are essential to prevent early rank decay, as previously shown in (Dong et al., 2021).

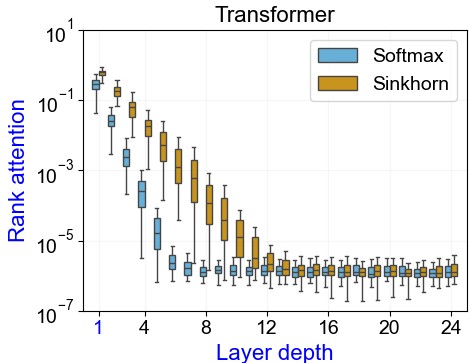

Figure 2: Rank collapse for the product of attention matrices as layer depth increases. Results obtained by training a Transformer on AG's news dataset (Zhang et al., 2015), see Section 4.1.

## 2   Preliminaries

We introduce the fundamental components of the self-attention mechanism, with particular focus on the doubly stochastic case, and review the Sinkhorn algorithm and its associated operator (Sinkhorn, 1964; Sinkhorn & Knopp, 1967), deferring the details of its iterative normalization procedure to Appendix A. We then recall the path decomposition argument proposed in (Dong et al., 2021), and the concept of residual that will be used both to obtain the norm bound in Section 3 and to measure rank decay in Section 4.

We start by briefly recalling the standard formulation of a self-attention head in Transformers (Vaswani et al., 2017). Consider an input matrix $X \in \mathbb{R}^{n \times d}$, where $n$ is the number of tokens in the sequence and $d$ the embedding dimension. Each token is associated with three projected representations: a *query*, a *key*, and a *value*. The attention matrix $P \in \mathbb{R}^{n \times n}$ is computed by measuring the similarity between queries and keys, producing weights that determine how strongly each token attends to the others. These weights are then used to compute a weighted aggregation of the value vectors, yielding the updated representation of each token.

In the classical setting of row-stochastic self-attention (Vaswani et al., 2017), the mapping $P$ is the softmax operator applied row-wise, thus normalizing each row to sum to one. In our case, $P$ is the doubly stochastic attention matrix obtained via the Sinkhorn operator, which iteratively normalizes both rows and columns to sum to one:

$$P = \text{Sinkhorn}\left( \frac{(XW_{Q,h} + \mathbf{1}b_{Q,h}^\top)(XW_{K,h} + \mathbf{1}b_{K,h}^\top)^\top}{\sqrt{d_{qk}}} \right), \tag{1}$$

with $W_{Q,h}, W_{K,h} \in \mathbb{R}^{d \times d_{qk}}$ query and key weight matrices, $b_{Q,h}^\top, b_{K,h}^\top \in \mathbb{R}^{d_{qk}}$ bias row vectors, and $\sqrt{d_{qk}}$ the scaling factor normalizing the score magnitudes.

In 1, Sinkhorn is obtained via a converging iterative algorithm (Sinkhorn, 1964; Sinkhorn & Knopp, 1967):

$$\text{Sinkhorn} : \mathbb{R}^{n \times n} \longrightarrow \text{DS}_{n \times n}$$

where $\text{DS}_{n \times n}$ denotes the space of doubly stochastic matrices, that is:

$$\text{DS}_{n \times n} := \{S \in \mathbb{R}^{n \times n} \mid S\mathbf{1} = \mathbf{1}, \ \mathbf{1}^\top S = \mathbf{1}^\top\},$$

with $\mathbf{1} \in \mathbb{R}^n$ denoting a column vector with all entries equal to 1. In Appendix A, we provide a description of the Sinkhorn algorithm and of its log domain implementation.

We define the *self-attention head operator* $\text{SA}_h$ acting on the input $X$ as follows:

$$\text{SA}_h(X) := PXW_{V,h} + \mathbf{1}b_{V,h}^\top, \tag{2}$$

where $W_{V,h} \in \mathbb{R}^{d \times d_v}$ is the value weight matrix and $b_{V,h}^\top \in \mathbb{R}^{d_v}$ is the bias row vector.

We next follow the path decomposition framework in (Dong et al., 2021) to write the general expression for a multi-head multi-layer network. We define the *self-attention network* SAN as a sequence of $L$ multi-head self-attention layers, each with $H$ heads. The output of each layer is obtained by concatenating the outputs of all its heads (along the last dimension) and linearly projecting them onto a subspace of appropriate size through a weight matrix $W_{O,h}$. The SAN output takes the form:

$$\text{SAN}(X) := \sum_{h_1,\ldots,h_L \in [H]^L} \left(P_{h_L}^L \cdots P_{h_1}^1\right) X \left(W_{h_1}^1 \cdots W_{h_L}^L\right) \tag{3}$$

where $[H] = (1, \ldots, H)$, $W_h^l = W_{V,h}^l {W_{O,h}^l}^\top$ with $\ell \in \{1, \ldots, L\}$, and $L$ the number of layers. In here, without loss of generality, we have discarded the bias term.

In (Dong et al., 2021), 3 is called a *path decomposition* for the self-attention network, the SAN being decomposed into a linear combination of *paths*. Each path corresponds to selecting a single attention head at each layer (or bypassing the layer via a skip connection), thereby forming an effective single-head network across layers.

From classical results on products of row-stochastic (Wolfowitz, 1963; Anthonisse & Tijms, 1977) and doubly stochastic matrices (Schwarz, 1980), it is known that, under sufficiently strong mixing (ergodic) conditions, such products converge to a rank-one matrix as the number of factors increases. In particular, products of doubly stochastic matrices converge to $\frac{1}{n}\mathbf{1}\mathbf{1}^\top$ (see (Schwarz, 1980)). Thus, we want to estimate how fast the product of doubly stochastic matrices $(P_{h_L}^L \cdots P_{h_1}^1)$ in 3 contributes to reducing the rank of the output matrix $\text{SAN}(X)$ as depth increases. To quantify this effect, we recall the notion of residual (Dong et al., 2021).

We define the *residual* matrix $\text{res}(X)$ of the input $X \in \mathbb{R}^{n \times d}$ as:

$$\text{res}(X) := X - \mathbf{1}x^\top, \qquad x := \frac{1}{n}X^\top\mathbf{1} \in \mathbb{R}^d \tag{4}$$

Here, $\mathbf{1}x^\top = \frac{1}{n}\mathbf{1}\mathbf{1}^\top X$ is the matrix obtained by uniform averaging across rows, and we recognize in $\frac{1}{n}\mathbf{1}\mathbf{1}^\top$ the rank-one limit of products of doubly stochastic matrices.

Analogously, for the output $\text{SAN}(X)$, we define:

$$\text{res}(\text{SAN}(X)) := \text{SAN}(X) - \mathbf{1}x_{\text{SAN}(X)}^\top, \quad x_{\text{SAN}(X)} := \frac{1}{n}\text{SAN}(X)^\top\mathbf{1} \tag{5}$$

We use the residual definition consistently across layers to relate $\text{res}(\text{SAN}(X))$ to $\text{res}(X)$ and track how intermediate representations evolve toward the rank-one limit.

With these preliminaries in mind, our goal is to study how products of attention matrices affect the rank of the representations produced by a self-attention network. In particular, we address the following questions:

1. How does the residual norm $\|\text{res}(\text{SAN}(X))\|_2$ decay as the number of layers increases when the self-attention matrices are normalized with Sinkhorn, and are therefore doubly stochastic? Equivalently, how quickly does the output of a self-attention network $\text{SAN}(X)$ approach a rank-one matrix under successive products of such attention matrices?

2. How does the implementation of attention normalization via the Sinkhorn algorithm, producing doubly stochastic attention matrices, affect the rank decay of products of attention matrices, and consequently of the self-attention output, compared to the standard row-stochastic softmax normalization used in Transformers?

In Section 3, we derive a theoretical bound on the residual norm $\|\text{res}(\text{SAN}(X))\|_2$ in terms of $\|\text{res}(X)\|_2$ when self-attention is normalized via Sinkhorn, yielding doubly stochastic matrices. We stress, however, that the expression of $\text{SAN}(X)$ in 3 corresponds to a pure self-attention network. A Transformer architecture is instead characterized by blocks composed of a self-attention layer followed by a feed-forward layer, with skip connections and layer normalizations in between. We will take into account the full Transformer architecture in our experiments in Section 4. In particular, in Section 4.1 we measure the rank decay of the product of attention matrices $\left(P_{h_L}^L \cdots P_{h_1}^1\right)$ in 3, while in Section 4.2 the one of the self-attention output $\text{SAN}(X)$.

## 3 Main theoretical results

We report here our main theoretical results, namely the norm bound for the rank decay of pure self-attention, without skip connections and feed-forward layers. We measure rank decay through the residual matrix and provide the formulas for a single-head single-layer $\text{SA}_h(X)$ in Theorem 1, and a multi-head multi-layer $\text{SAN}(X)$ in Theorem 2. We briefly outline details about the proof in Section 3.1, referring the reader to the appendices for the full details.

We first state the result for a single-head single-layer $\text{SA}_h(X)$, whose proof is provided in Appendix D.

**Theorem 1** (Single-head, single-layer). *For a single-head single-layer $\text{SA}_h(X)$, the residual satisfies:*

$$\boxed{\|\text{res}(\text{SA}_h(X))\|_2 \leq \frac{\lambda\,\beta}{\sqrt{n^3 d_{qk}}}\|\text{res}(X)\|_2^3}$$

*where $0 < \lambda \leq 1$, $n$ is the sequence length, $d_{qk}$ is the query and key embedding dimension, and $\|W_Q W_K^\top\|_2 \|W_V\|_2 \leq \beta$ for some $\beta > 0$.*

We then state the result for the multi-head multi-layer $\text{SAN}(X)$, which relies mainly on Theorem 1. Its proof is reported in Appendix D together with the single-head multi-layer and the multi-head single-layer cases.

**Theorem 2** (Multi-head, multi-layer). *For any multi-head $\text{SAN}(X)$ consisting of $H$ heads and $L$ layers, with $\|W_Q W_K^\top\|_2 \|W_h\|_2 \leq \beta$ for every head $h \in \{1, \ldots, H\}$ and every layer $\ell \in \{1, \ldots, L\}$, the residual satisfies:*

$$\boxed{\|\text{res}(\text{SAN}(X))\|_2 \leq \left(\frac{\lambda\,\beta H}{\sqrt{n^3 d_{qk}}}\right)^{\frac{3^L-1}{2}}\|\text{res}(X)\|_2^{3^L}}$$

As observed in (Dong et al., 2021) for the row-stochastic softmax case, pure self-attention converges to a rank-one matrix doubly exponentially with depth also when the attention is normalized to be doubly stochastic via Sinkhorn. We recall that the classical results for products of stochastic matrices show convergence to a rank-one matrix at an exponential rate as the number of factors increases (Anthonisse & Tijms, 1977), rather than doubly exponential. The cubic rate of convergence arises from the fact that the attention matrix $P$ depends on the input $X$ and is then applied to it, see 6 in Section 3.1. Indeed, while classical results require mixing conditions on products of independent stochastic matrices, in Transformers mixing arises from the dependence of the stochastic matrices on the input and on one another, leading to a faster convergence (see also Appendix G).

With respect to the norm bound in (Dong et al., 2021), we highlight that we employ $\ell_2$, which is a proper norm, while they employ a composition of $\ell_1$ and $\ell_\infty$ norms which does not satisfy triangle inequality. See Appendix E for the relationship between $\ell_2$ and their $\ell_{1,\infty}$. We further note that, due to the approximation in Lemma 1 in Section 3.1, the factor $\sqrt{n^3}$ naturally appears in the denominator of the coefficient multiplying $\mathrm{res}(X)$ in the norm bound. If, after training, the quantity $\|W_Q W_K^\top\|_2 \|W_h\|_2$ remains close to its initialization scale (which is typically of order $O(1)$ under common initialization schemes), the coefficient $\frac{\lambda \beta H}{\sqrt{n^3 d_{qk}}}$ is strictly smaller than 1, without requiring additional assumptions on the self-attention matrix.

For completeness, in Appendix D we also report the bound for the multi-head multi-layer SAN($X$) with skip connections on. This theoretically shows how skip connections help slowing down rank decay, as previously found in (Dong et al., 2021) and in accordance with our experimental results in Section 4.

### 3.1 Proof Sketch

We now give some details about the proof of Theorem 1, from which also the proof of Theorem 2 follows.

**The projection operator $Q$.** First of all, let's define the linear map:

$$Q : \mathbb{R}^{n \times n} \longrightarrow \mathrm{TDS}_{n \times n}, \qquad Q(Y) = Y - \frac{1}{n}\mathbf{1}\mathbf{1}^\top Y - \frac{1}{n}Y\mathbf{1}\mathbf{1}^\top + \frac{1}{n^2}\mathbf{1}\mathbf{1}^\top Y \mathbf{1}\mathbf{1}^\top,$$

$Q$ is the orthogonal projection onto the space $\mathrm{TDS}_{n \times n}$, which represents the tangent space to the doubly stochastic matrix space:

$$\mathrm{TDS}_{n \times n} := \{H \in \mathbb{R}^{n \times n} \mid H\mathbf{1} = 0, \ \mathbf{1}^\top H = 0\}$$

see Appendix B for more details.

**Lemma 1.** *The orthogonal projection $Q$ represents a first-order approximation of the Sinkhorn operator at zero:*

$$\mathrm{Sinkhorn}(tY) = \frac{1}{n}\mathbf{1}\mathbf{1}^\top + \frac{t}{n^2}Q(Y) + o(t), \qquad for\ Y \in \mathbb{R}^{n \times n}$$

In Appendix C we show that:

$$(C \circ R)(tY) = \frac{1}{n}\mathbf{1}\mathbf{1}^\top + \frac{t}{n^2}Q(Y) + o(t)$$

where $C(Y)$ and $R(Y)$ denote the column and row softmax normalizations of a matrix $Y$, respectively. Since the Sinkhorn operator consists of a composition of such operators and $Q$ is a projection operator, i.e. $Q^2 = Q$, the stated first-order approximation follows.

**Proposition 1.** *Let the notation be as above. We have:*

$$\|Q(Y)\|_2 \le \|Q(Y)\|_F \le \lambda \sqrt{\mathrm{rank}(Y)} \, \|Y\|_2$$

*where $0 < \lambda \le 1$, while $\|\cdot\|_2$ and $\|\cdot\|_F$ denote respectively the $\ell_2$ and the Frobenius norm in the space of $n \times n$ matrices.*

This is a consequence of the orthogonal decomposition with respect to the Frobenius norm: $Y = Q(Y) + (I - Q)(Y)$, and of the relationship between the $\ell_2$ and the Frobenius norm for matrices, see Appendix B for more details.

**Proof details.** The key to the proof of Theorem 1 is to exploit the shift-invariance of the Sinkhorn operator $P$ with respect to constant column or row matrices, so that the self-attention mechanism for a single-head single-layer $\mathrm{SA}_h$ is rewritten as:

$$\mathrm{SA}_h = P(A)XW_V = \mathrm{Sinkhorn}(A)XW_V, \quad \text{with} \quad A := R\frac{W_{QK}}{\sqrt{d_{qk}}}R^\top, \quad R := \mathrm{res}(X), \tag{6}$$

where we have omitted the head index $h$ and $W_{QK} = W_Q W_K^\top$.

Then, we employ the projection operator $Q$ in Lemma 1 to approximate $P(A)$:

$$P(A)X \approx \left[\frac{1}{n}\mathbf{1}\mathbf{1}^\top + \frac{1}{n^2}Q(A)\right]X = \mathbf{1}x^\top + \frac{1}{n^2}Q(A)(X - \mathbf{1}x^\top) = \mathbf{1}x^\top + \frac{1}{n^2}Q(A)R, \qquad (7)$$

where $\frac{1}{n}\mathbf{1}\mathbf{1}^\top X = \mathbf{1}x^\top$ and $Q(A)\mathbf{1}x^\top = 0$.

We multiply 7 by $W_V$ on both sides and we get:

$$P(A)XW_V - \mathbf{1}x^\top W_V \approx \frac{1}{n^2}Q(A)RW_V \qquad (8)$$

By explicitly computing the residual matrix of $\mathrm{SA}_h(X)$, we obtain:

$$\mathrm{res}(\mathrm{SA}_h(X)) = P(A)XW_V - \mathbf{1}x^\top W_V \qquad (9)$$

Then, we substitute 9 in 8 and taking the $\ell_2$ norm gives:

$$\|\mathrm{res}(\mathrm{SA}_h(X))\|_2 \le \frac{1}{n^2}\|Q(A)\|_2 \|W_V\|_2 \|\mathrm{res}(X)\|_2$$

By using the estimate on $Q(A)$ given by Proposition 1, we obtain the norm bound result in Theorem 1.

## 4 Experiments

In this section, we are interested in measuring the effective rank decay in a Transformer architecture, extending the analysis beyond the pure self-attention case, while taking into account the effect of skip connections and feed-forward layers. Following (Dong et al., 2021), we consider four different settings: a pure self-attention network $\mathrm{SAN}(X)$; a $\mathrm{SAN}(X)$ with skip connections; a $\mathrm{SAN}(X)$ with feed-forward layers; and a Transformer with both skip connections and feed-forward layers.

Models are trained using the full Transformer architecture, including both skip connections and feed-forward layers. The rank analysis is then performed at inference time under the four configurations described above, without retraining. To this end, we modify the forward pass by selectively disabling skip connections and/or bypassing the feed-forward layers. In particular, skip connections are removed by avoiding the addition of the residual input, while feed-forward layers are bypassed by omitting their application.

Our goal is to compare rank decay in a trained architecture when attention is normalized with softmax (row-stochastic) and when it is normalized with Sinkhorn (doubly stochastic). In all experiments, we use existing Transformer architectures and train them from scratch, i.e., starting from random initialization of all model parameters rather than from pretrained weights, by replacing the softmax operator with the Sinkhorn algorithm. We employ the code provided in (Sander et al., 2022)[1], which implements Sinkhorn normalization in the log domain for numerical stability (see Appendix A). Additional experimental details are reported in Appendix F.

**Datasets and tasks.** We introduce the datasets and tasks used in our experiments, which include both text and image classification. For text classification, we train an encoder-only Transformer on the AG's News dataset (Zhang et al., 2015). The architecture consists of a self-attention encoder followed by a max pooling over the sequence length and a linear classification layer to predict the article category. The task is to classify each news article into one of four categories: "World", "Sports", "Business", and "Science/Technology". For image classification, we train an encoder-only Vision Transformer (ViT) (Dosovitskiy et al., 2020) on the MNIST (Deng, 2012) and Cats and Dogs (Kaggle, 2013) datasets. The architecture consists of a self-attention encoder followed by a linear classification layer to predict the image class. The task is to classify images into the ten digits classes for MNIST and the two classes for Cats and Dogs.

---

[1] https://github.com/michaelsdr/sinkformers

### 4.1 Rank decay along attention paths

We study how quickly products of attention matrices approach rank one as the number of factors increases. Figures 3a to 3c show the normalized residual with respect to the best rank-one approximation for products of attention matrices sampled from trained Transformer models. The $x$-axis reports the depth of the product, i.e., the number of matrices multiplied, while the $y$-axis shows the normalized spectral norm of the residual.

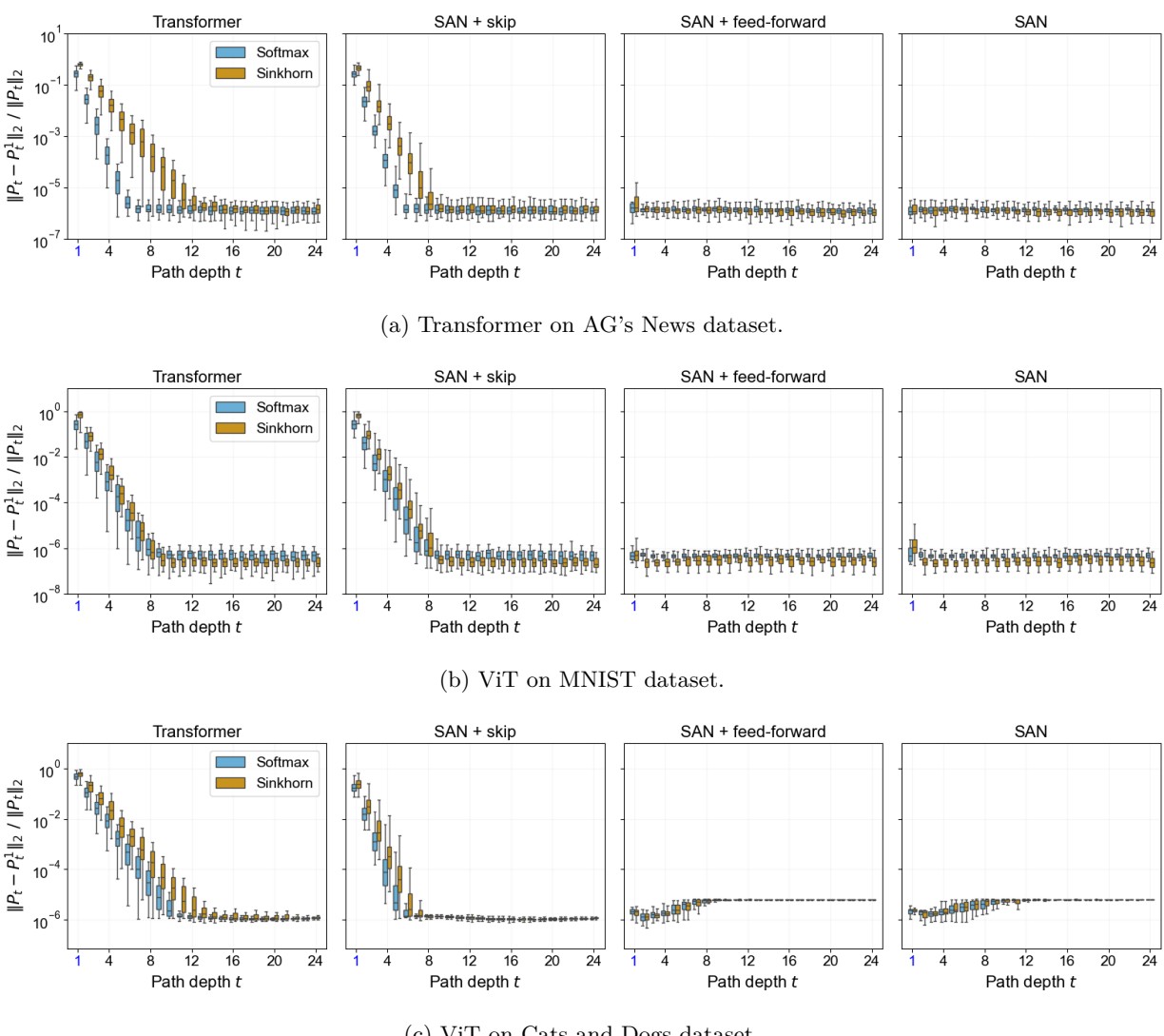

(a) Transformer on AG's News dataset.

(b) ViT on MNIST dataset.

(c) ViT on Cats and Dogs dataset.

Figure 3: Normalized spectral norm of $\mathrm{res}(P_t)$ in 11 as a function of path depth $t$, with the $y$-axis on a logarithmic scale. For each depth, results are estimated from 100 sampled attention paths in trained Transformer architectures. The central line indicates the median, while the box spans the interquartile range, and the whiskers extend to non-outlier values. Lower values indicate rank closer to one.

Across datasets and architectures, we observe that Sinkhorn doubly stochastic normalization mitigates rank collapse compared to row-stochastic softmax normalization in the early layers, before rank collapse becomes too severe for a comparison to be meaningful. We note that this occurs when skip connections are present, otherwise rank is low to start with and no comparison is then possible. This behavior was first reported in (Dong et al., 2021), which shows that skip connections are essential to prevent rank collapse in self-attention layers, while without them the rank of the hidden representations decays immediately (see also Figures 4a to 4c).

In the path decomposition argument of (Dong et al., 2021), skip connections are interpreted as omitting factors in the product of attention matrices $\left(P_{h_L}^L \cdots P_{h_1}^1\right)$ appearing in 3. Each path corresponds to a sequence of attention heads across layers, and a skip connection effectively bypasses a layer, reducing the number of matrices in the product. This provides an intuitive explanation for why skip connections slow down rank decay: they shorten the sequence of stochastic matrices being multiplied. Since repeated products of stochastic matrices converge to a rank-one matrix as the number of factors increases (Wolfowitz, 1963; Anthonisse & Tijms, 1977), introducing skips interrupts this process and helps preserve rank.

We now describe how the products of attention matrices used in these plots are constructed. For each depth $t \in \{1, \ldots, T\}$, we estimate how close a product of $t$ randomly sampled attention matrices from the trained Transformer is to being rank one. Let $P_{h_\ell,b}^\ell \in \mathbb{R}^{n \times n}$ denote the attention matrix at layer $\ell \in \{1, \ldots, L\}$, head $h \in \{1, \ldots, H\}$, and sample $b \in \{1, \ldots, \beta\}$, where $\beta$ is the batch size and $n$ is the sequence length.

We follow (Dong et al., 2021) and construct a random attention path of depth $t$ as follows:

1. Sample $t$ distinct layers $\{\ell_1, \ldots, \ell_t\}$ uniformly without replacement, and sort them so that $\ell_1 < \cdots < \ell_t$.

2. For each selected layer $\ell_i \in \{\ell_1, \ldots, \ell_t\}$, sample one head $h$ and one batch index $b$ uniformly.

The chained attention product at depth $t$ is:

$$P_t = \prod_{i=1}^{t} P_{h_{\ell_i},b}^{\ell_i} \quad \in \mathbb{R}^{n \times n},\tag{10}$$

where matrices are multiplied in increasing layer order.

To quantify how close $P_t$ is to rank one, we compute its best rank-one approximation via truncated singular value decomposition at the first-order:

$$P_t^1 = \sigma_1 u_1 v_1^\top,$$

where $\sigma_1$ is the largest singular value of $P_t$ and $u_1, v_1$ are the corresponding singular vectors.

The resulting residual matrix of $P_t$ is approximated as:

$$\text{res}(P_t) = P_t - P_t^1,\tag{11}$$

and we measure its normalized spectral norm as $\frac{\|\text{res}(P_t)\|_2}{\|P_t\|_2}$. To obtain an empirical estimate, we repeat this sampling procedure 100 times for each depth $t$. Results are reported in Figures 3a to 3c.

To further substantiate the claim that Sinkhorn preserves rank better than softmax in the early layers and in the presence of skip connections, we complement the visual results in Figures 3a to 3c with a direct empirical count. Specifically, after sampling 100 paths per depth, we compute the percentage of paths for which Sinkhorn achieves better rank preservation than softmax, measured by the normalized residual $\frac{\|\text{res}(P_t)\|_2}{\|P_t\|_2}$, and report the average over the first 8 layers. We focus in particular on the full Transformer, which is the architecture we are ultimately interested in. In this setting, Sinkhorn outperforms softmax on 98.6% of paths on AG's News, 78.8% on Cats and Dogs, and 65.5% on MNIST, further supporting our claim.

In Appendix G, we study rank decay in products of randomly generated stochastic matrices and observe that, in this unstructured setting, the difference between softmax and Sinkhorn normalization disappears. This indicates that doubly stochasticity alone does not generally improve rank preservation when matrices are independent and random. In contrast, when attention matrices are extracted from trained Transformers, Sinkhorn consistently yields better rank preservation. This discrepancy suggests that the improvement observed in models trained with Sinkhorn cannot be attributed solely to the per-layer attention normalization, but also depends on how attention matrices interact across layers. Indeed, attention matrices in a Transformer are not independent but correlated with one another (see additional details in Appendix G). Understanding how such correlations interact with doubly stochastic constraints to affect rank preservation is an interesting direction for future work.

### 4.2 Rank decay of the self-attention output along layers

Let $\mathrm{SAN}(X) \in \mathbb{R}^{L \times \beta \times n \times d}$ denote the output of the self-attention network, where $L$ is the number of layers, $\beta$ the batch size, $n$ the number of tokens, and $d$ the embedding dimension. For each layer $\ell \in \{1, \ldots, L\}$ and batch element $b \in \{1, \ldots, \beta\}$, we denote by $\mathrm{SAN}(X)_{\ell,b} \in \mathbb{R}^{n \times d}$ the matrix of token representations at that layer and for that data sample. To approximate the limit rank-one matrix of identical representations, we follow (Dong et al., 2021) and we average on the rows of $\mathrm{SAN}(X)_{\ell,b}$:

$$x_{\mathrm{SAN}(X)_{\ell,b}}^\top = \frac{1}{n} \mathbf{1}^\top \mathrm{SAN}(X)_{\ell,b} \ \in \ \mathbb{R}^d$$

The residual of $\mathrm{SAN}(X)_{\ell,b}$ is then computed as:

$$\mathrm{res}(\mathrm{SAN}(X)_{\ell,b}) = \mathrm{SAN}(X)_{\ell,b} - \mathbf{1}x_{\mathrm{SAN}(X)_{\ell,b}}^\top, \tag{12}$$

and we measure its normalized spectral norm as $\frac{\|\mathrm{res}(\mathrm{SAN}(X)_{\ell,b})\|_2}{\|\mathrm{SAN}(X)_{\ell,b}\|_2}$.

For each layer $\ell$, we plot the batch mean and standard deviation. In Figure 4a we show the results for AG's news dataset, while the other two Figures 4b and 4c report MNIST (Deng, 2012) and Cats and Dogs (Kaggle, 2013) results.

In accordance with Figures 3a to 3c, which report the behavior of chained attention products, Figures 4a to 4c show that rank is preserved when skip connections are present. In contrast, when skip connections are removed, rank decays from the very beginning, consistently with the findings of Dong et al. (2021). We note that in Figures 3a to 3c we plot the rank of attention matrices starting from the first layer, while in Figures 4a to 4c we start from the input embeddings before the first layer, which have higher rank and then rapidly collapse.

When skip connections are there, a clear advantage of Sinkhorn normalization over softmax is observed only in Figure 4a, where the rank of $\mathrm{SAN}(X)$ is better preserved, consistently with Figure 3a. In contrast, for Figures 4b and 4c, softmax and Sinkhorn normalization exhibit very similar behavior in terms of $\mathrm{res}(\mathrm{SAN}(X))$ across layers, despite the advantage of Sinkhorn observed in the rank of attention matrix products in Figures 3b and 3c. Indeed, with respect to the path attention product defined in 10, the output of $\mathrm{SAN}(X)$ is not solely determined by this attention product, but also by its multiplication with the input $X$ and the products of matrices $\left(W_{h_1}^1 \cdots W_{h_L}^L\right)$, see 3. We emphasize that the most direct and informative way to compare the effect of softmax and Sinkhorn normalization on rank collapse across layers is to analyze the attention matrices themselves, as done in Section 4.1.

## 5 Limitations

We highlight two directions for future refinement:

1. **Training protocol.** Following the methodology of Dong et al. (2021), in Section 4 we train full Transformers and disable feed-forward layers only at inference to isolate the pure self-attention behavior, with or without skip connections. While the experimental results agree with our theoretical bounds on self-attention's rank decay, an interesting direction for future work is to train $\mathrm{SAN}(X)$ models from scratch.

2. **Tightness of bounds.** The theoretical bounds obtained for Sinkhorn, similar to the ones obtained for softmax in Dong et al. (2021), relay on estimates that could potentially be tightened.

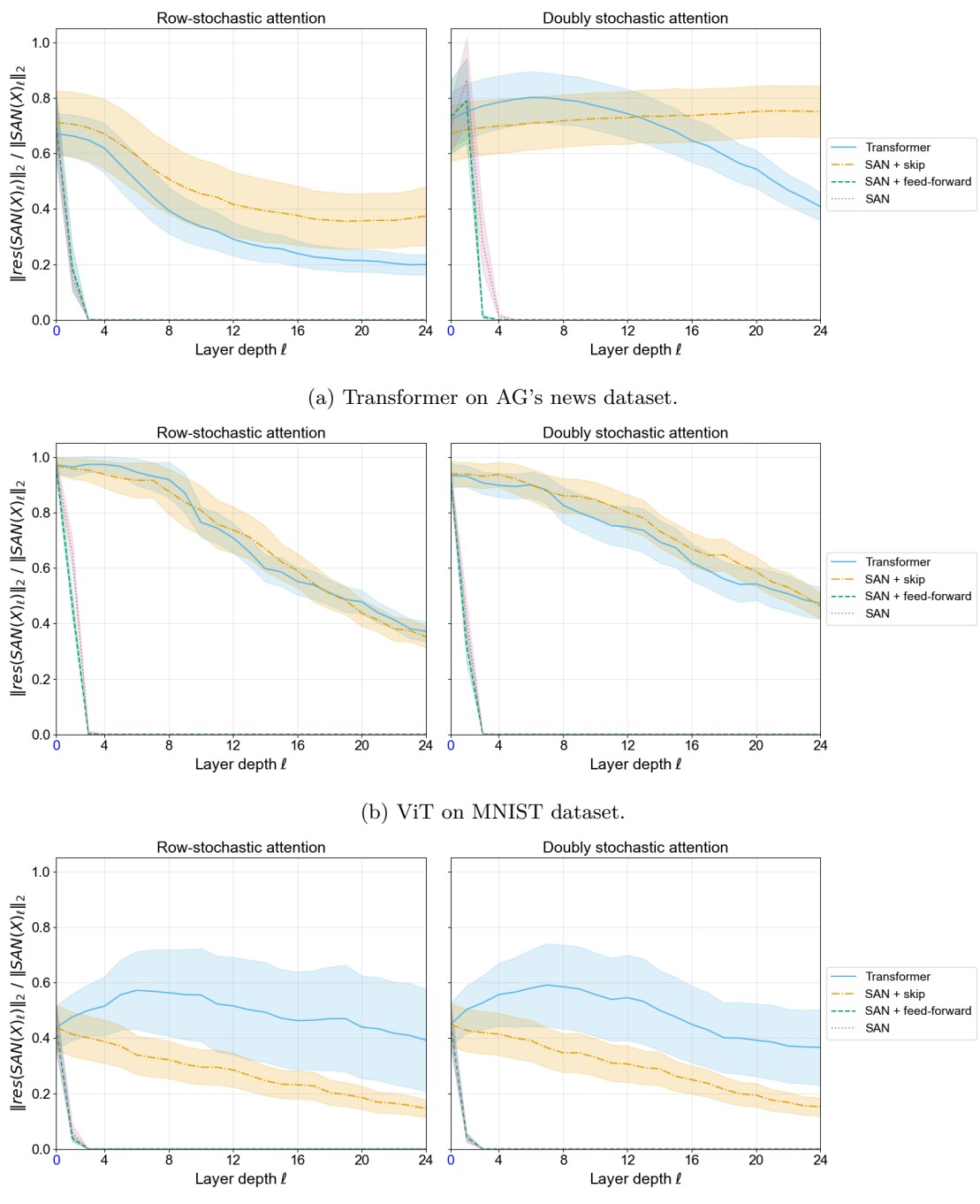

(a) Transformer on AG's news dataset.

(b) ViT on MNIST dataset.

(c) ViT on Cats and Dogs dataset.

Figure 4: Normalized spectral norm of res($\text{SAN}(X)_\ell$) in 12 as a function of layer depth $\ell$. Results show the mean over the batch, with shaded regions indicating one standard deviation. We use $\text{SAN}(X)$ in the y-axis label to denote all four settings: a pure self-attention network, a $\text{SAN}(X)$ with skip connections, a $\text{SAN}(X)$ with feed-forward layers, and a Transformer with both skip connections and feed-forward layers. The value at $\ell = 0$ corresponds to the input embeddings before the first self-attention layer.

## 6 Conclusion

We study the phenomenon of rank collapse in a Transformer architecture, when the attention matrix is normalized to be doubly stochastic using the Sinkhorn algorithm. While previous work has analyzed rank collapse for standard row-stochastic softmax attention, the behavior of doubly stochastic normalization has not yet been characterized.

From a theoretical perspective, we derive norm bounds for the decay of the residual matrix in a pure self-attention network without skip connections and feed-forward layers. Our analysis shows that, similarly to the softmax case studied in (Dong et al., 2021), pure self-attention with Sinkhorn normalization converges to a rank-one matrix doubly exponentially with depth. The proof relies on a first-order approximation of the Sinkhorn operator and employs the spectral norm, which provides a proper norm-based estimate for the decay.

From an empirical perspective, we evaluate rank decay in trained Transformer architectures on both natural language and vision tasks. Our experiments show that doubly stochastic normalization mitigates rank collapse compared to standard softmax normalization. This effect is most evident in the early layers, before rank collapse becomes severe, and in the presence of skip connections, which are known to be essential for preventing rank decay (Dong et al., 2021).

An interesting direction for future work concerns the stability properties of attention mechanisms. Recent work Nair (2025) shows that the softmax operator is $\frac{1}{2}$-Lipschitz with respect to any norm and uses this result to refine the global Lipschitz constant for scaled cosine similarity attention (Qi et al., 2023), as it has been previously shown that the Lipschitz constant for standard row-stochastic self-attention is infinite with respect to any norm (Kim et al., 2020). Comparing the Lipschitz properties of Sinkhorn-normalized doubly stochastic attention with those of softmax could provide further insight into the robustness and optimization behavior of Transformer architectures, since larger Lipschitz constants typically imply higher sensitivity to input perturbations and more challenging training dynamics (Cranko et al., 2018).

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

# Appendices

## A  Sinkhorn algorithm and its implementation

In (Sinkhorn, 1964), Sinkhorn shows that for each positive matrix $Y \in \mathbb{R}_{>0}^{n \times n}$ there is a unique doubly stochastic matrix $S = D_1 Y D_2$, i.e., satisfying $\sum_{j=1}^{n} S_{ij} = 1$ and $\sum_{i=1}^{n} S_{ij} = 1$, where $D_1$ and $D_2$ are diagonal matrices with positive main diagonals.

The doubly stochastic matrix $S$ is obtained as the limit of the sequence of matrices generated by alternatively normalizing the rows and columns of $Y$. The row $\mathcal{R}$ and column $\mathcal{C}$ normalization operators are defined as:

$$\mathcal{R}(Y) = \operatorname{diag}\left(\frac{1}{\sum_{j=1}^{n} Y_{ij}}\right)_{i=1}^{n} Y, \qquad \mathcal{C}(Y) = Y \operatorname{diag}\left(\frac{1}{\sum_{i=1}^{n} Y_{ij}}\right)_{j=1}^{n}$$

$$(\mathcal{R}(Y))_{ij} = \frac{Y_{ij}}{\sum_{k=1}^{n} Y_{ik}}, \qquad (\mathcal{C}(Y))_{ij} = \frac{Y_{ij}}{\sum_{k=1}^{n} Y_{kj}}, \tag{13}$$

and the sequence of matrices $\{\mathcal{Y}^{(k)}\}_{k \geq 0}$ is:

$$\mathcal{Y}^{(0)} = Y, \quad \mathcal{Y}^{(2k+1)} = \mathcal{R}(\mathcal{Y}^{(2k)}), \quad \mathcal{Y}^{(2k+2)} = \mathcal{C}(\mathcal{Y}^{(2k+1)}), \quad \text{with } \lim_{k \to \infty} \mathcal{Y}^{(k)} = S$$

In (Sinkhorn & Knopp, 1967), this result is generalized to nonnegative matrices $Y$.

In our implementation, we follow (Sander et al., 2022)[2] and employ Sinkhorn algorithm to solve the entropy-regularized optimal transport problem between queries and keys, where the transport cost is given by the unnormalized attention scores. More details about how the optimal transport problem reduces to a matrix scaling problem can be found in (Vialard, 2019). The solution to the optimal transport problem is obtained by alternately normalizing the rows and columns of the transport matrix until convergence. Convergence is estimated empirically and occurs when successive Sinkhorn iterations change the matrix by less than a predefined threshold. To improve numerical stability, the normalization is performed in the log domain. Convergence of the procedure is preserved because the log and exp functions are bijective.

## B  The projection operator $Q$

For the reader's convenience we detail some properties of the projection operator:

$$Q : \mathbb{R}^{n \times n} \longrightarrow \mathrm{TDS}_{n \times n}, \qquad Q(Y) = Y - \frac{1}{n} \mathbf{1}\mathbf{1}^\top Y - \frac{1}{n} Y \mathbf{1}\mathbf{1}^\top + \frac{1}{n^2} \mathbf{1}\mathbf{1}^\top Y \mathbf{1}\mathbf{1}^\top, \tag{14}$$

where $\mathbb{R}^{n \times n}$ are the $n \times n$ real matrices and:

$$\mathrm{TDS}_{n \times n} := \{H \in \mathbb{R}^{n \times n} \mid H\mathbf{1} = 0, \ \mathbf{1}^\top H = 0\}$$

---

[2] https://github.com/michaelsdr/sinkformers

**Proposition 2.** *The linear map $Q$ is an orthogonal projection with respect to the Frobenius inner product in the space of $n \times n$ matrices, which is defined by:*

$$\langle A, B \rangle_F := \sum_{i,j=1}^{n} a_{ij} b_{ij}, \qquad A = (a_{ij}),\ B = (b_{ij}) \in \mathbb{R}^{n \times n}$$

*Proof.* We first show that $Q$ is well-defined, i.e., $Q(Y) \in \mathrm{TDS}_{n \times n}$. Indeed:

$$Q(Y)\mathbf{1} \ = Y\mathbf{1} - \tfrac{1}{n}\mathbf{1}\mathbf{1}^\top Y \mathbf{1} - \tfrac{1}{n} Y \mathbf{1}\mathbf{1}^\top \mathbf{1} + \tfrac{1}{n^2}\mathbf{1}\mathbf{1}^\top Y \mathbf{1}\mathbf{1}^\top \mathbf{1} =$$

$$= Y\mathbf{1} - \tfrac{1}{n}\mathbf{1}\mathbf{1}^\top Y \mathbf{1} - \tfrac{1}{n} Y \mathbf{1} n + \tfrac{1}{n^2}\mathbf{1}\mathbf{1}^\top Y \mathbf{1} n = 0$$

Similarly, one shows that $\mathbf{1}^\top Q(Y) = 0$, hence $Q(Y) \in \mathrm{TDS}_{n \times n}$.

Next, we show that $Q$ is symmetric with respect to $\langle \cdot, \cdot \rangle_F$. It is enough to verify that:

$$\langle Q(E_{ij}), E_{\alpha\beta} \rangle_F = \langle E_{ij}, Q(E_{\alpha\beta}) \rangle_F$$

that is,

$$Q(E_{ij})_{\alpha\beta} = Q(E_{\alpha\beta})_{ij},$$

where $E_{ij}$ denotes an elementary matrix in $\mathbb{R}^{n \times n}$, with $i, j, \alpha, \beta \in \{1, \ldots, n\}$.

We compute:

$$Q(E_{ij}) = E_{ij} - \frac{1}{n} \sum_{k=1}^{n} E_{kj} - \frac{1}{n} \sum_{l=1}^{n} E_{il} + \frac{1}{n^2} \sum_{k,l=1}^{n} E_{kl}$$

Hence:

$$Q(E_{ij})_{\alpha\beta} = \delta_{i\alpha}\delta_{j\beta} - \frac{1}{n}\delta_{j\beta} - \frac{1}{n}\delta_{i\alpha} + \frac{1}{n^2}$$

Interchanging the roles of $i, j$ with $\alpha, \beta$ we see immediately that $Q(E_{ij})_{\alpha\beta} = Q(E_{\alpha\beta})_{ij}$.

We leave to the reader the straightforward check that $Q$ is a projection operator, i.e., $Q^2 = Q$, and that $Q(Y)$ is orthogonal to $(I - Q)(Y)$ with respect to the Frobenius inner product, which is an immediate consequence of $Q$ being symmetric. $\qquad \square$

**Proposition 3.** *Let the notation be as above. We have:*

$$\|Q(Y)\|_2 \le \|Q(Y)\|_F \le \lambda \sqrt{\mathrm{rank}(Y)} \, \|Y\|_2$$

*where $0 < \lambda \le 1$, while $\|\cdot\|_2$ and $\|\cdot\|_F$ denote respectively the $\ell_2$ and the Frobenius norm in the space of $n \times n$ matrices:*

$$\|A\|_2 := \sup\{\|Ax\|_2 : x \in \mathbb{R}^n \text{ such that } \|x\|_2 \le 1\} \qquad \|A\|_F := \sqrt{\sum_{i,j=1}^{n} |a_{ij}|^2}$$

*Proof.* We write $Y$ as its orthogonal decomposition with respect to the Frobenius inner product:

$$Y = Q(Y) + (I - Q)(Y)$$

Since $\langle Q(Y), (I - Q)(Y) \rangle_F = 0$, then:

$$\|Y\|_F^2 = \|Q(Y)\|_F^2 + \|(I - Q)(Y)\|_F^2$$

Since $Y$ is not necessarily in $\mathrm{TDS}_{n \times n}$, we have:

$$\|(I - Q)(Y)\|_F \ge 0,$$

with equality if and only if $Y \in \text{TDS}_{n \times n}$, and strict inequality otherwise.

Consequently, there exists a constant $0 < \lambda \leq 1$ such that:

$$\|Q(Y)\|_F \leq \lambda \|Y\|_F \tag{15}$$

Since the Frobenius norm is related to the 2-norm of a matrix as:

$$\|Y\|_F \leq \sqrt{\text{rank}(Y)} \, \|Y\|_2$$

Applying this norm bound to the contraction inequality in 15, we get:

$$\|Q(Y)\|_F \leq \lambda \|Y\|_F \leq \lambda \sqrt{\text{rank}(Y)} \, \|Y\|_2$$

$\square$

## C  The projection operator $Q$ as Sinkhorn Approximation

We now show that the operator $Q$, as defined in Appendix B, represents a first-order approximation of the Sinkhorn operator.

Sinkhorn operator is obtained via successive applications of row-softmax $R : \mathbb{R}^{n \times n} \longrightarrow \mathbb{R}^{n \times n}$ and column-softmax $C : \mathbb{R}^{n \times n} \longrightarrow \mathbb{R}^{n \times n}$ operators (see Appendix A):

$$(R(Y))_{ij} = \frac{e^{y_{ij}}}{\sum_{k=1}^{n} e^{y_{ik}}}, \qquad (C(Z))_{ij} = \frac{e^{z_{ij}}}{\sum_{k=1}^{n} e^{z_{kj}}}$$

where $Y, Z \in \mathbb{R}^{n \times n}$. It is then enough to show that $Q$ represents a first-order approximation of the composition of $C$ and $R$.

**Proposition 4.** *Let $F = C \circ R : \mathbb{R}^{n \times n} \longrightarrow \mathbb{R}^{n \times n}$ with $R$ row-softmax and $C$ column-softmax operators. Define $Q : \mathbb{R}^{n \times n} \longrightarrow \mathbb{R}^{n \times n}$, $Q(Y) = Y - \frac{1}{n}\mathbf{1}\mathbf{1}^\top Y - \frac{1}{n}Y\mathbf{1}\mathbf{1}^\top + \frac{1}{n^2}\mathbf{1}\mathbf{1}^\top Y \mathbf{1}\mathbf{1}^\top$. Then:*

$$F(tX) = \frac{1}{n}\mathbf{1}\mathbf{1}^\top + \frac{t}{n^2}Q(X) + o(t) \tag{16}$$

*Proof.* Let $s : \mathbb{R}^n \longrightarrow \mathbb{R}^n$ denote the softmax function. Its Jacobian at $u \in \mathbb{R}^n$ is given by:

$$J_s(u) = \text{diag}(s(u)) - s(u)s(u)^\top$$

In particular, evaluating at $u = 0$, we obtain:

$$J_s(0) = \frac{1}{n}I - \frac{1}{n^2}\mathbf{1}\mathbf{1}^\top = \frac{1}{n}U,$$

where $U := I - \frac{1}{n}\mathbf{1}\mathbf{1}^\top$ is the orthogonal projection in $\mathbb{R}^n$ onto $\mathbf{1}^\perp$.

For a one-parameter perturbation $u(t) = 0 + tv$ with $v \in \mathbb{R}^n$, we have:

$$s(tv) = s(0) + tJ_s(0)v + o(t) = \frac{1}{n}\mathbf{1} + \frac{t}{n}Uv + o(t). \tag{17}$$

For $Y \in \mathbb{R}^{n \times n}$, define:

$$r_i(Y) := \begin{pmatrix} y_{i1} \\ \vdots \\ y_{in} \end{pmatrix} \in \mathbb{R}^n, \qquad c_j(Y) := \begin{pmatrix} y_{1j} \\ \vdots \\ y_{nj} \end{pmatrix} \in \mathbb{R}^n,$$

From 17, it follows that:

$$r_i(R(tX)) = s(t\, r_i(X)) = \frac{1}{n}\mathbf{1} + \frac{t}{n}U\, r_i(X) + o(t),$$

which in matrix form can be written as:

$$R(tX) = \frac{1}{n}\mathbf{1}\mathbf{1}^\top + \frac{t}{n}XU + o(t).$$

We now apply the column-softmax operator to $Z(t) := R(tX)$. We can write:

$$Z(t) = Z_0 + t\Delta + o(t),$$

where $Z_0 = \frac{1}{n}\mathbf{1}\mathbf{1}^\top$ and $\Delta = \frac{1}{n}XU$.

Since each column of $Z_0$ equals $\frac{1}{n}\mathbf{1}$, and since adding a constant to a column does not change softmax, for each column $j$, we have that:

$$c_j(C(Z(t))) = \frac{1}{n}\mathbf{1} + \frac{t}{n}U\,c_j(\Delta) + o(t).$$

Written in matrix form, this means that:

$$C(Z(t)) = \frac{1}{n}\mathbf{1}\mathbf{1}^\top + \frac{t}{n}U\Delta + o(t).$$

We are now ready to compute:

$$F(tX) = C(R(tX)) = C(Z(t)) = \frac{1}{n}\mathbf{1}\mathbf{1}^\top + \frac{t}{n}U\Delta + o(t) = \frac{1}{n}\mathbf{1}\mathbf{1}^\top + \frac{t}{n}U\left(\frac{1}{n}XU\right) + o(t).$$

Therefore:

$$F(tX) = \frac{1}{n}\mathbf{1}\mathbf{1}^\top + \frac{t}{n^2}UXU + o(t) = \frac{1}{n}\mathbf{1}\mathbf{1}^\top + \frac{t}{n^2}Q(X) + o(t),$$

where we have used the fact that $Q(X) = UXU$. $\qquad\square$

The error term $o(t)$ in Proposition 4 involves higher-order softmax derivatives whose denominators scale with powers of $n$ greater than 2, making them negligible.

## D  Proofs of the main theoretical results

We report here the proof of the rank decay theorems provided in Section 3.

**Theorem 3** (Single-head, single-layer)**.** *For a single-head single-layer* $\mathrm{SA}_h(X)$*, the residual satisfies:*

$$\boxed{\|\mathrm{res}(\mathrm{SA}_h(X))\|_2 \le \frac{\lambda\,\beta}{\sqrt{n^3 d_{qk}}}\|\mathrm{res}(X)\|_2^3}$$

*where* $0 < \lambda \le 1$*,* $n$ *the sequence length,* $d_{qk}$ *the query and key embedding dimension, and* $\|W_Q W_K^\top\|_2\|W_V\|_2 \le \beta$ *for some* $\beta > 0$*.*

*Proof.* From 1, the unscaled attention scores for a single head (omitting the head index $h$ for simplicity) are computed as:

$$(XW_Q + \mathbf{1}b_Q^\top)(XW_K + \mathbf{1}b_K^\top)^\top \tag{18}$$

with $W_Q, W_K \in \mathbb{R}^{d\times d_{qk}}$ query and key weight matrices.

Since the Sinkhorn operator produces a doubly stochastic matrix, all terms in 18 that provide a constant contribution across rows or columns can be safely removed because of shift-invariance. This yields:

$$P((XW_Q + \mathbf{1}b_Q^\top)(XW_K + \mathbf{1}b_K^\top)^\top) = P(XW_{QK}X^\top), \tag{19}$$

where $W_{QK} = W_Q W_K^\top$ and $P$ denotes the Sinkhorn operator.

We use the shorthand notation $R := \text{res}(X)$, where $\text{res}(X)$ is the residual defined in 4, so that $X = \mathbf{1}x^\top + R$. Substituting this into 19 gives:

$$P\left((\mathbf{1}x^\top + R)\frac{W_{QK}}{\sqrt{d_{qk}}}(\mathbf{1}x^\top + R)^\top\right) = P\left(\frac{x^\top W_{QK} x}{\sqrt{d_{qk}}}\mathbf{1}\mathbf{1}^\top + R\frac{W_{QK}}{\sqrt{d_{qk}}}x\mathbf{1}^\top + \mathbf{1}x^\top\frac{W_{QK}}{\sqrt{d_{qk}}}R^\top + R\frac{W_{QK}}{\sqrt{d_{qk}}}R^\top\right)$$

$$= P(A), \qquad A := R\frac{W_{QK}}{\sqrt{d_{qk}}}R^\top \tag{20}$$

where the last equality follows from the shift-invariance of the Sinkhorn operator.

We now approximate $P(A)$ by Proposition 4 in Appendix C:

$$P(A)X \approx \left[\frac{1}{n}\mathbf{1}\mathbf{1}^\top + \frac{1}{n^2}Q(A)\right]X = \mathbf{1}x^\top + \frac{1}{n^2}Q(A)(X - \mathbf{1}x^\top) = \mathbf{1}x^\top + \frac{1}{n^2}Q(A)R \tag{21}$$

Since $\frac{1}{n}\mathbf{1}\mathbf{1}^\top X = \mathbf{1}x^\top$ and $Q(A)\mathbf{1}x^\top = 0$.

Multiplying 21 by $W_V$, and writing $P$ in place of $P(A)$ to ease the notation, we obtain:

$$PXW_V - \mathbf{1}x^\top W_V \approx \frac{1}{n^2}Q(A)\,RW_V$$

Applying the submultiplicativity of the spectral norm yields:

$$\|PXW_V - \mathbf{1}x^\top W_V\|_2 \le \frac{1}{n^2}\|Q(A)\|_2\,\|R\|_2\,\|W_V\|_2 \tag{22}$$

By Proposition 1, we have:

$$\|Q(A)\|_2 \le \lambda\sqrt{\text{rank}(A)}\,\|A\|_2 \tag{23}$$

Combining the inequalities in 22 and 23 gives the bound:

$$\|PXW_V - \mathbf{1}x^\top W_V\|_2 \le \frac{\lambda\sqrt{\text{rank}(A)}}{n^2}\|A\|_2\,\|R\|_2\,\|W_V\|_2 \tag{24}$$

Since $\text{rank}(A)$ is at most equal to $n$, we can rewrite 24 as:

$$\|PXW_V - \mathbf{1}x^\top W_V\|_2 \le \frac{\lambda}{n^{3/2}}\|A\|_2\,\|R\|_2\,\|W_V\|_2 \tag{25}$$

Since by definition $A = \frac{RW_{QK}R^\top}{\sqrt{d_{qk}}}$, by submultiplicativity of the spectral norm we obtain:

$$\|A\|_2 = \left\|\frac{RW_{QK}R^\top}{\sqrt{d_{qk}}}\right\|_2 \le \frac{1}{\sqrt{d_{qk}}}\|R\|_2^2\|W_{QK}\|_2 \tag{26}$$

Substituting the bound in 26 into the previous inequality in 25 yields:

$$\|PXW_V - \mathbf{1}x^\top W_V\|_2 \le \frac{\lambda}{\sqrt{n^3 d_{qk}}}\|W_{QK}\|_2\|W_V\|_2\|R\|_2^3 \tag{27}$$

Following the definition of residual in Section 2, we compute the residual of the single-head single-layer output $\text{SA}_h(X)$ as:

$$\text{res}(\text{SA}_h(X)) := \text{SA}_h(X) - \mathbf{1}x_{\text{SA}_h(X)}^\top, \quad x_{\text{SA}_h(X)} := \frac{1}{n}\text{SA}_h(X)^\top\mathbf{1} \tag{28}$$

Substituting the $\mathrm{SA}_h(X)$ expression from 2, and omitting the head index $h$ for simplicity, we obtain:

$$\mathrm{res}(\mathrm{SA}_h(X)) = PXW_V - \frac{1}{n}\mathbf{1}\mathbf{1}^\top PXW_V, \tag{29}$$

where the bias term is omitted without loss of generality.

Since $P$ is doubly stochastic, $\mathbf{1}^\top P = \mathbf{1}^\top$, and therefore:

$$\mathrm{res}(\mathrm{SA}_h(X)) = PXW_V - \mathbf{1}x^\top W_V$$

Taking the spectral norm and using the previous bound in 27 gives:

$$\|\mathrm{res}(\mathrm{SA}_h(X))\|_2 = \|PXW_V - \mathbf{1}x^\top W_V\|_2 \leq \frac{\lambda}{\sqrt{n^3 d_{qk}}}\|W_{QK}\|_2\|W_V\|_2\|\mathrm{res}(X)\|_2^3$$

$\square$

**Theorem 4** (Single-head, multi-layer)**.** *For any single-head* $\mathrm{SAN}_h(X)$ *consisting of* $L$ *layers, with* $\|W_Q W_K^\top\|_2\|W_V\|_2 \leq \beta$ *for every layer* $\ell \in \{1, \ldots, L\}$*, the residual satisfies:*

$$\boxed{\|\mathrm{res}(\mathrm{SAN}_h(X))\|_2 \leq \left(\frac{\lambda\beta}{\sqrt{n^3 d_{qk}}}\right)^{\frac{3^L-1}{2}} \|\mathrm{res}(X)\|_2^{3^L}}$$

*and hence converges to zero at a doubly exponential rate in the depth* $L$*.*

*Proof.* We consider $L$ layers of the form $X^l = \mathrm{SA}_h{}^l(X^{l-1})$ and we unfold the recursion backwards from the last layer to the first one, by using the bound on $\mathrm{SA}_h(X)$ in Theorem 3:

$$\|\mathrm{res}(\mathrm{SAN}(X))\|_2 = \|\mathrm{res}(X)^L\|_2 \leq \frac{\lambda\beta}{\sqrt{n^3 d_{qk}}}\|\mathrm{res}(X)^{L-1}\|_2^3 \leq \frac{\lambda\beta}{\sqrt{n^3 d_{qk}}}\left(\frac{\lambda\beta}{\sqrt{n^3 d_{qk}}}\|\mathrm{res}(X)^{L-2}\|_2^3\right)^3 \tag{30}$$

$$= \frac{\lambda\beta}{\sqrt{n^3 d_{qk}}}\left(\frac{\lambda\beta}{\sqrt{n^3 d_{qk}}}\right)^3\|\mathrm{res}(X)^{L-2}\|_2^{3^2} \leq \prod_{l=1}^{L}\left(\frac{\lambda\beta}{\sqrt{n^3 d_{qk}}}\right)^{3^{l-1}}\|\mathrm{res}(X)\|_2^{3^L} \tag{31}$$

$$= \left(\frac{\lambda\beta}{\sqrt{n^3 d_{qk}}}\right)^{\frac{3^L-1}{2}}\|\mathrm{res}(X)\|_2^{3^L} \tag{32}$$

where:

$$\prod_{l=1}^{L}\left(\frac{\lambda\beta}{\sqrt{n^3 d_{qk}}}\right)^{3^{l-1}} = \left(\frac{\lambda\beta}{\sqrt{n^3 d_{qk}}}\right)^{\sum_{l=1}^{L}3^{l-1}} = \left(\frac{\lambda\beta}{\sqrt{n^3 d_{qk}}}\right)^{\frac{3^L-1}{2}}$$

by the geometric series sum. $\square$

**Theorem 5** (Multi-head, single-layer)**.** *For any single-layer* $\mathrm{SA}(X)$ *consisting of* $H$ *heads, with* $\|W_Q W_K^\top\|_2\|W_h\|_2 \leq \beta$ *for every head* $h \in \{1, \ldots, H\}$*, the residual satisfies:*

$$\boxed{\|\mathrm{res}(\mathrm{SA}(X))\|_2 \leq \frac{\lambda\beta H}{\sqrt{n^3 d_{qk}}}\|\mathrm{res}(X)\|_2^3}$$

*Proof.* The output of a multi-head attention layer is: $\mathrm{SA}(X) = \sum_{h\in[H]}\mathrm{SA}_h(X) = \sum_{h\in[H]}P_h X W_h$, where $W_h = W_{h,V}W_{O,h}^\top$. The residual of the attention output $\mathrm{SA}(X)$ is then:

$$\mathrm{res}(\mathrm{SA}(X)) = \mathrm{res}\Big(\sum_{h\in[H]}\mathrm{SA}_h(X)\Big) = \sum_{h\in[H]}\mathrm{SA}_h(X) - \mathbf{1}x_{\mathrm{SA}(X)}^\top$$

where:

$$x_{\text{SA}(X)}^\top = \frac{1}{n}\mathbf{1}^\top\Big(\sum_{h\in[H]}P_h X W_h\Big) = \frac{1}{n}\sum_{h\in[H]}\mathbf{1}^\top P_h X W_h = \frac{1}{n}\sum_{h\in[H]}\mathbf{1}^\top X W_h = \sum_{h\in[H]}x^\top W_h$$

since $P$ is doubly stochastic and $\mathbf{1}^\top P_h = \mathbf{1}^\top$.

Thus the residual becomes:

$$\text{res}(\text{SA}(X)) = \sum_{h\in[H]}\big(\text{SA}_h(X) - \mathbf{1}x^\top W_h\big)$$

For the subadditivity of the spectral norm we have:

$$\|\sum_{h\in[H]}\big(\text{SA}_h(X) - \mathbf{1}x^\top W_h\big)\|_2 \leq \sum_{h\in[H]}\|\big(\text{SA}_h(X) - \mathbf{1}x^\top W_h\big)\|_2 = \sum_{h\in[H]}\|\text{res}(\text{SA}_h(X))\|_2$$

Since from Theorem 3 each $\text{SA}_h(X))$ satisifes $\|\text{res}(\text{SA}_h(X))\|_2 \leq \frac{\lambda\beta}{\sqrt{n^3 d_{qk}}}\|\text{res}(X)\|_2^3$, then:

$$\|\text{res}(\text{SA}(X))\|_2 \leq \frac{\lambda\beta H}{\sqrt{n^3 d_{qk}}}\|\text{res}(X)\|_2^3$$

$\square$

**Theorem 6** (Multi-head, multi-layer). *For any multi-head* $\text{SAN}(X)$ *consisting of $H$ heads and $L$ layers, with* $\|W_Q W_K^\top\|_2\|W_h\|_2 \leq \beta$ *for every head $h \in \{1,\ldots,H\}$ and every layer $\ell \in \{1,\ldots,L\}$, the residual satisfies:*

$$\boxed{\|\text{res}(\text{SAN}(X))\|_2 \leq \left(\frac{\lambda\beta H}{\sqrt{n^3 d_{qk}}}\right)^{\frac{3^L-1}{2}}\|\text{res}(X)\|_2^{3^L}}$$

*Proof.* The proof follows recursively as in Theorem 4, this time using the multi-head residual formula obtained in Theorem 5. $\square$

**Theorem 7** (Multi-head, multi-layer SAN with skip connections). *For any multi-head* $\text{SAN}(X)$ *consisting of $H$ heads and $L$ layers, with* $\|W_Q W_K^\top\|_2\|W_h\|_2 \leq \beta$ *for every head $h \in \{1,\ldots,H\}$ and every layer $\ell \in \{1,\ldots,L\}$, and skip connections on, the residual satisfies:*

$$\boxed{\|\text{res}(\text{SAN}(X) + X)\|_2 \leq \max_{0\leq l\leq L}\left(\frac{\lambda\beta H}{\sqrt{n^3 d_{qk}}}\right)^{\frac{3^l-1}{2}} 2H^{3^l(L-l)}\|\text{res}(X)\|_2^{3^l}}$$

*where the maximum over $l$ replaces the full-depth factor* $\left(\frac{\lambda\beta H}{\sqrt{n^3 d_{qk}}}\right)^{\frac{3^L-1}{2}}$ *in Theorem 6, mitigating rank collapse.*

*Proof.* The residual of a single-head single-layer $\text{SA}_h(X)$ with skip connections on is given by:

$$\text{res}(SA_h(X) + X)$$

Following the definition of residual in Section 2:

$$\text{res}(SA_h(X) + X) = (SA_h(X) + X) - \mathbf{1}\, x_{(SA_h(X)+X)}^T,$$

$$x_{(SA_h(X)+X)} = \frac{1}{n}(SA_h(X) + X)^T \mathbf{1},$$

$$\mathbf{1}x_{(SA_h(X)+X)}^T = \frac{1}{n}\mathbf{1}\mathbf{1}^T(SA_h(X) + X)$$

Since $SA_h(X) = PXW_V$, we have:

$$\mathbf{1}x^T_{(SA_h(X)+X)} = \frac{1}{n}\mathbf{1}\mathbf{1}^TPXW_V + \frac{1}{n}\mathbf{1}\mathbf{1}^TX$$
$$= \mathbf{1}x^TW_V + \mathbf{1}x^T,$$

where we recall that $\frac{1}{n}\mathbf{1}\mathbf{1}^\top X = \mathbf{1}x^\top$.

Therefore:

$$\text{res}(SA_h(X) + X) = PXW_V - \mathbf{1}x^TW_V + X - \mathbf{1}x^T$$
$$= \text{res}(SA_h(X)) + \text{res}(X)$$

Hence:

$$\|\text{res}(SA_h(X) + X)\|_2 = \|\text{res}(SA_h(X)) + \text{res}(X)\|_2$$
$$\leq \|\text{res}(SA_h(X))\|_2 + \|\text{res}(X)\|_2$$

Given the previous result on $\|\text{res}(SA_h(X))\|_2$ in Theorem 1:

$$\|\text{res}(SA_h(X) + X)\|_2 \leq \frac{\lambda\beta}{\sqrt{n^3 d_{qk}}}\|\text{res}(X)\|_2^3 + \|\text{res}(X)\|_2$$

We now follow Dong et al. (2021) and unfold the recursion backwards to obtain a multi-layer bound. We have that:

$$\|\text{res}(\mathbf{X}^L)\|_2 \leq \frac{\lambda\beta}{\sqrt{n^3 d_{qk}}}\|\text{res}(\mathbf{X}^{L-1})\|_2^3 + \|\text{res}(\mathbf{X}^{L-1})\|_2$$
$$\leq 2\max\left(\frac{\lambda\beta}{\sqrt{n^3 d_{qk}}}\|\text{res}(\mathbf{X}^{L-1})\|_2^3, \ \|\text{res}(\mathbf{X}^{L-1})\|_2\right) \tag{33}$$

where, since both terms are nonnegative, we use the elementary scalar inequality $a + b \leq 2\max(a, b)$.

We unroll this bound across layers to express it in terms of $\text{res}(\mathbf{X})$. At each step of unrolling, the max is one of the two terms in 33. Unrolling through $L$ layers thus corresponds to tracing a path from root to leaf in a complete binary tree of depth $L$, where the final bound is determined by the maximum over all such paths. Each path has the form:

$$\left(\frac{\lambda\beta}{\sqrt{n^3 d_{qk}}}\right)^{\frac{3^l-1}{2}} 2^{\,3^l(L-l)}\,\|\text{res}(\mathbf{X})\|_2^{3^l},$$

where $l$ indicates the number of times the term $\frac{\lambda\beta}{\sqrt{n^3 d_{qk}}}\|\text{res}(\mathbf{X}^{L-k})\|_2^3$ is chosen as the max. The residual bound is the maximum amongst such paths:

$$\|\text{res}(\mathbf{X}^L)\|_2 \leq \max_{0\leq l\leq L}\left(\frac{\lambda\beta}{\sqrt{n^3 d_{qk}}}\right)^{\frac{3^l-1}{2}} 2^{\,3^l(L-l)}\|\text{res}(\mathbf{X})\|_2^{3^l}$$

We now apply this bound to $H$ heads. For a single-layer this is:

$$\|\operatorname{res}(\mathrm{SA}(\mathbf{X}))\|_2 \leq \frac{\lambda\beta}{\sqrt{n^3 d_{qk}}}\|\operatorname{res}(\mathbf{X})\|_2^3 + H\|\operatorname{res}(\mathbf{X})\|_2$$

While for a multi-layer this is finally:

$$\|\operatorname{res}(\mathbf{X}^L)\|_2 \leq \max_{0 \leq l \leq L}\left(\frac{\lambda\beta}{\sqrt{n^3 d_{qk}}}\right)^{\frac{3^l-1}{2}}(2H)^{3^l(L-l)}\|\operatorname{res}(\mathbf{X})\|_2^{3^l},$$

which concludes the proof. □

# E    Relationship between $\ell_2$ and $\ell_{1,\infty}$

We want to derive the relationship between the $\ell_2$ norm and the $\ell_{1,\infty}$ norm defined in (Dong et al., 2021).

For any matrix $A \in \mathbb{R}^{n \times n}$, the $\ell_2$ norm is defined as:

$$\|A\|_2 := \sup\{\|Ax\|_2 : x \in \mathbb{R}^n \text{ such that } \|x\|_2 \leq 1\},$$

while the $\ell_1$ and $\ell_\infty$ norms of a matrix are defined as:

$$\|A\|_1 := \max_{1 \leq j \leq n} \sum_{i=1}^{n} |a_{ij}|, \qquad \|A\|_\infty := \max_{1 \leq i \leq n} \sum_{j=1}^{n} |a_{ij}|.$$

Following (Dong et al., 2021), the $\ell_{1,\infty}$ norm is defined as:

$$\|A\|_{1,\infty} := \sqrt{\|A\|_1 \|A\|_\infty}$$

We now show that the spectral norm $\ell_2$ is bounded by this quantity.

First, recall that the spectral norm can be expressed as the square root of the largest eigenvalue of $A^\top A$, namely:

$$\|A\|_2 = \sqrt{\rho(A^\top A)}, \tag{34}$$

where $\rho(\cdot)$ denotes the spectral radius.

For any matrix $B$, the spectral radius satisfies $\rho(B) \leq \|B\|_1$.

Squaring 34 and applying this inequality to $B = A^\top A$ gives:

$$\|A\|_2^2 = \rho(A^\top A) \leq \|A^\top A\|_1$$

Next, using the submultiplicativity of the $\ell_1$ norm, we obtain:

$$\|A\|_2^2 \leq \|A^\top A\|_1 \leq \|A^\top\|_1 \|A\|_1$$

Since $\|A^\top\|_1 = \|A\|_\infty$, it follows that:

$$\|A\|_2^2 \leq \|A^\top A\|_1 \leq \|A\|_\infty \|A\|_1$$

Taking the square root on both sides yields:

$$\|A\|_2 \leq \sqrt{\|A\|_1 \|A\|_\infty} = \|A\|_{1,\infty}$$

## F    Experimental details

We report the main implementation details for the text and image classification experiments in Section 4. Across all experiments, we compare standard row-stochastic attention, obtained by applying the softmax operator row-wise, with doubly stochastic attention, obtained by iteratively applying Sinkhorn algorithm (see Appendix A). Empirically, we observe that the attention matrices become effectively doubly stochastic after 50 iterations of the Sinkhorn algorithm, and this value is used throughout the experiments. All models are trained with Adam optimizer (Kingma & Ba, 2014) and cross-entropy loss on the number of classes.

**Text classification.**    For text classification, we employ the AG's News dataset (Zhang et al., 2015), which contains news articles grouped into four categories: "World", "Sports", "Business", and "Science/Technology". Each input example is a sequence of tokens representing the article text. The tokenized sequences are processed by a Transformer encoder composed of stacked blocks, each consisting of a self-attention layer followed by a feed-forward layer, with skip connections and layer normalizations in between. The final prediction is obtained by applying max pooling over the sequence length, followed by a linear classification layer to predict the article category. All hyperparameters are grouped in Table 1.

Table 1: Hyperparameters for the text classification experiment. The query and key dimension $d_{qk}$, and the value dimension $d_v$, are derived from the model dimension $d_{\text{model}}$ as $d_{qk} = d_v = d_{\text{model}}/H$.

| Dataset | AG's News |
|---|---|
| Maximum sequence length | 128 |
| Model dimension | 128 |
| Feed-forward dimension | 512 |
| Attention heads $H$ | 4 |
| Number of layers | 24 |
| Batch size | 32 |
| Learning rate | $5 \times 10^{-5}$ |
| Dropout | 0.3 |
| Epochs | 20 |

**Image classification.**    For image classification, we consider two datasets: MNIST (Deng, 2012) and Cats and Dogs (Kaggle, 2013). MNIST contains grayscale images of handwritten digits, while Cats and Dogs is a binary classification dataset of RGB images depicting cats and dogs. Each image is divided into non-overlapping patches, which are flattened and linearly projected into token embeddings of dimension equal to the model dimension (see Table 2). The resulting sequence of tokens is processed by a Vision Transformer (ViT) encoder composed of stacked blocks, each consisting of a self-attention layer followed by a feed-forward network, with residual connections and layer normalization. A special learnable classification token (CLS) is prepended to the sequence and interacts with all patch tokens through self-attention, enabling it to aggregate global information about the image. The final prediction is obtained by feeding the representation of this CLS token into a linear classification layer to predict the image category. All hyperparameters are grouped in Table 2.

## G    Rank decay in random stochastic matrix products

To further investigate rank decay when comparing row-stochastic and doubly stochastic normalization, we measure the residual in 11 for products of randomly generated matrices that are subsequently normalized using either softmax or Sinkhorn. The goal of this experiment is to assess whether the behavior observed in Figures 3a to 3c for attention paths in trained Transformer architectures is a more general property of stochastic matrices. In particular, we test whether products of randomly sampled stochastic matrices exhibit stronger rank decay in the row-stochastic (softmax) than in the doubly stochastic (Sinkhorn) case. The resulting plot is shown in Figure 5.

Table 2: Hyperparameters for the image classification experiments. The query and key dimension $d_{qk}$, and the value dimension $d_v$, are derived from the model dimension $d_{\text{model}}$ as $d_{qk} = d_v = d_{\text{model}}/H$.

|                        | MNIST              | Cats and Dogs      |
|------------------------|--------------------|--------------------|
| Image size             | $28 \times 28$     | $224 \times 224$   |
| Patch size             | 4                  | 16                 |
| Model dimension        | 128                | 128                |
| Feed-forward dimension | 512                | 512                |
| Attention heads $H$    | 4                  | 4                  |
| Number of layers       | 24                 | 24                 |
| Batch size             | 100                | 32                 |
| Learning rate          | $2 \times 10^{-2}$ | $3 \times 10^{-5}$ |
| Dropout                | 0                  | 0                  |
| Epochs                 | 50                 | 50                 |

From Figure 5, we observe that when there is no correlation between the stochastic matrices in the product $P_t$ defined in 10, softmax row-stochastic and Sinkhorn doubly stochastic normalizations exhibit essentially the same rank decay as the number of factors (depth) increases. The same behavior is observed when a random subset of factors in the product is skipped to mimic the presence of skip connections. In particular, this effect is simulated by replacing half of the matrices with the identity.

This suggests that the advantage of Sinkhorn doubly stochastic normalization observed in Figures 3a to 3c may not be a generic property of arbitrary stochastic matrices. Rather, it may be linked to the fact that attention matrices along the path in a Transformer are correlated with one another, with the attention matrix $P_{h_{\ell+1}}^{\ell+1}$ at layer $\ell + 1$ taking as input the output $P_{h_\ell}^\ell X W_{h_\ell}^\ell$ of the previous layer $\ell$, see 3. A deeper understanding of how Sinkhorn doubly stochastic attention matrices interact, in comparison to softmax row-stochastic ones, is a challenging problem that we leave for future work.

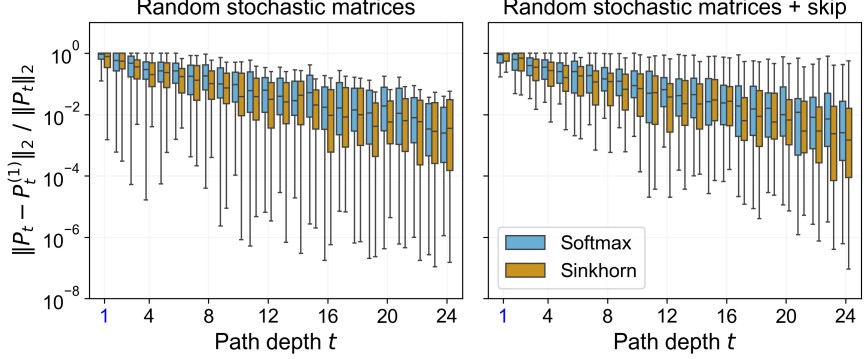

Figure 5: Normalized spectral norm of $\text{res}(P_t)$ in 11 as a function of path depth $t$. The attention matrices in $P_t$ are randomly generated rather than extracted from a trained Transformer. For each depth, results are estimated from 100 sampled attention paths. Lower values indicate rank closer to one.

We also observe that the rank of products of randomly generated stochastic matrices decays more slowly than in the case of attention matrices extracted from a trained Transformer (Figures 3a to 3c). This is consistent with classical results showing that products of stochastic matrices converge to a rank-one matrix at an exponential rate as the number of factors increases (Anthonisse & Tijms, 1977), whereas in Transformers the decay is doubly exponential with depth, with a cubic rate. As already mentioned in Section 3, the cubic rate of convergence arises from the fact that the attention matrix $P$ depends on the input $X$ and is applied to it. Since the input to the attention matrix at layer $\ell + 1$ is the output $P_{h_\ell}^\ell X W_{h_\ell}^\ell$ of the previous layer $\ell$, this dependency on $X$ propagates across layers and leads to a cascading effect that amplifies rank decay as depth increases. The attention matrices become progressively closer to rank-one as their inputs become

increasingly low-rank. While the intuition that stochastic matrices drive convergence still applies, these interactions result in a significantly faster rank collapse.

