# OpenReview forum: "Sinkhorn doubly stochastic attention rank decay analysis"
_TMLR — Under review for TMLR_

### Review · Reviewer_uwHV · 2026-05-31

**Summary Of Contributions:**

This work presents an analysis of the self-attention mechanism that is central to the transformer architecture. In particular, the analysis focuses on analyzing the rank of the conventional row-stochastic self-attention matrices and the effect of Sinkhorn normalization on the rate of rank collapse of these matrices.

The included theoretical analysis demonstrates that pure self-attention converges doubly exponentially to a rank-one matrix even under Sinkhorn normalization. Coupled with the experiments, this result suggests that Sinkhorn normalization yields more favorable constants in the normalization bound. Additionally, the empirical experiments seek to characterize the rank decay for pure self-attention, self-attention with feed-forward layers, self-attention with skip connections and the full transformer layer (self-attention + skip connections + feed-forward layers). Sinkhorn normalization shows the biggest improvement over row-stochastic normalization for the cases with skip connections. This may in part be due to the rapid rate decay in the other cases.

The motivations and objectives of this work are clearly set out and fulfilled. The theoretical analyses focused on the simpler pure self-attention setting, seeking to bound the residual and characterize the rank decay. They then proceed to empirical experiments to extend the analysis to more realistic settings and ablate the various components of the transformer layer, in a bid to isolate the elements responsible for rank preservation in transformers.

However, there are two key concerns that should be addressed.

1.	The discussion of the results in Fig. 3 makes taller claims than those borne out by the figures. Specifically, only the results on the AG’s News dataset show a demonstrable advantage of Sinkhorn normalization over row-stochastic normalization. This contrasts with the statement (on page 8): “Across datasets and architectures, we observe that Sinkhorn doubly stochastic normalization mitigates rank collapse compared to row-stochastic softmax normalization.”
2.	With the skip connections providing such an effective buffer against rank decay, this begs the question as to the efficacy or need for Sinkhorn normalization in practical transformer architectures.

**Audience:**

Yes

**Audience Explanation:**

Transformers are prevalent in modern-day ML arhcitectures, so analyses to better understand the performance of the underlying self-attention mechanism is likely to be of interest to many ML theorists and practitioners.

**Broader Impact Concerns:**

I do not have any broader impact concerns with this work.

**Claims And Evidence:**

Yes

**Claims Explanation:**

The motivations and objectives of this work are clearly set out and fulfilled. The theoretical analyses focused on the simpler pure self-attention setting, seeking to bound the residual and characterize the rank decay. They then proceed to empirical experiments to extend the analysis to more realistic settings and ablate the various components of the transformer layer, in a bid to isolate the elements responsible for rank preservation in transformers.

However, there are two key concerns that should be addressed.

1.	The discussion of the results in Fig. 3 makes taller claims than those borne out by the figures. Specifically, only the results on the AG’s News dataset show a demonstrable advantage of Sinkhorn normalization over row-stochastic normalization. This contrasts with the statement (on page 8): “Across datasets and architectures, we observe that Sinkhorn doubly stochastic normalization mitigates rank collapse compared to row-stochastic softmax normalization.”
2.	With the skip connections providing such an effective buffer against rank decay, this begs the question as to the efficacy or need for Sinkhorn normalization in practical transformer architectures.

**Requested Changes:**

**Critical changes:**

1.	The results in the two rightmost plots in Fig. 3 should be discussed. They appear to show that the attention matrices were low-rank to begin with. Why is this the case? And if so, are these experiments useful for analyzing rank decay?
2.	It is unclear what the authors mean by “the improved rank preservation observed in Transformers trained with doubly stochastic attention from the existence of correlations between attention matrices across layers” (on page 9).
3.	Why are the $\beta$ values derived on page 11?
4.	The authors claim that the relationships between the curves in Fig. 4 do not match Fig. 3. Pursuant to my previous comment, this claim needs substantiation.

**Suggested changes:**

5.	Relabeling the axis for Figure 2 would improve clarity as at that stage of the manuscript, it is not clear what the mathematical quantity on the y-axis represents.

---

> ### Author Response · Authors · 2026-06-10
>
> We thank the reviewer for the useful and insightful comments. We provide point-by-point responses in the following and upload a revised version of the manuscript with changes highlighted in blue.
>
> (1) Taller claims about Fig. 3:
>
> We agree with the reviewer that the sentence needs to be more specific and have revised it accordingly. The case in which Sinkhorn evidently mitigates rank collapse is when skip connections are present, otherwise rank collapse occurs from the beginning, making comparison impossible, and in early layers, after which both normalizations are severely affected by rank collapse. We have now modified accordingly the statements in the paper. To further substantiate that Sinkhorn mitigates rank collapse in this regime, we now report the mean percentage of paths for which Sinkhorn outperforms softmax in rank preservation over the first 8 layers: for the full Transformer, this is 98.6\% for AG’s News dataset, but also 78.8\% for Cats and Dogs and 65.5 \% for MNIST. These numbers complement the visual plots in Fig. 3 and confirm that the advantage of Sinkhorn holds consistently across datasets and architectures.
>
> (2) Skip connections vs. Sinkhorn:
>
> We fully agree that skip connections are essential for mitigating rank decay, as previously established in the literature (https://arxiv.org/pdf/2103.03404). Motivated by growing evidence that doubly stochastic attention improves performance, our aim was to show that, within a standard residual Transformer architecture, doubly stochastic (Sinkhorn) normalization further reduces rank collapse relative to conventional row-stochastic softmax normalization. We have now clarified this better in the text.
>
> (3) Low-rank without skip connections:
>
> The initial rank collapse observed for the attention matrices in Fig. 3 when skip connections are removed is consistent with the corresponding rank behavior of hidden states shown in Fig. 4. A similar instantaneous decay in the rank of hidden states without skip connections has been reported in prior work (https://arxiv.org/pdf/2103.03404) and is not the focus of our paper. While the underlying cause of this collapse is not addressed here, our main interest is the setting with residual connections, where rank has not yet fully collapsed and differences between normalization schemes can still be meaningfully assessed. We included the no-skip results only for transparency, and we have clarified this point in the revised manuscript. We can remove these plots, upon
> Referee's suggestion, to improve clarity.
>
> (4) Correlations in Transformer:
>
> We analyze rank decay using products of attention matrices extracted from Transformers trained with either softmax (row‑stochastic) or Sinkhorn (doubly stochastic), and observe consistently better rank preservation with Sinkhorn. When we repeat the experiment with randomly generated (hence uncorrelated) attention matrices normalized to be row‑ or doubly stochastic, overall rank decay is much smaller and softmax and Sinkhorn behave similarly. This discrepancy may arise because, in Transformers, attention matrices along a path are correlated with one another. Precisely characterizing how inter-layer correlations interact with doubly stochastic normalization to yield stronger rank preservation remains an open question. We have now clarified this better in the main text.
>
> (5) $\beta$ sentence on page 11:
>
> We apologize for the confusing word "obtain'' in the sentence "For each layer $\ell$, we obtain $\beta$ values and we plot the batch mean and standard deviation'' on page 11. The values of $\beta$ are not derived. What we meant is that, since the batch has size $\beta$ (i.e., $\beta$ samples), we compute the normalized residual for every batch element, and we plot the mean and standard deviation over these $\beta$ values. We have now simply modified the sentence to: "For each layer $\ell$, we plot the batch mean and standard deviation''.
>
> (6) Curves mismatch Fig. 4:
>
> What we intended to emphasize is that the attention matrix rank provides a more direct measure of the effect of Sinkhorn versus softmax normalization, whereas the rank of hidden states is influenced by additional components of the Transformer architecture. We agree that phrasing this in terms of a "relationship mismatch'' between curves in Figs. 3 and 4 was misleading, and we have therefore removed this sentence in the revised manuscript.
>
> (7) Relabeling Fig. 2:
>
> We agree and have now changed the y-axis to be more clear to the reader.

---

### Review · Reviewer_Po22 · 2026-06-10

**Summary Of Contributions:**

This paper characterized the decay rate of Sinkhorn doubly stochastic attention matrices.

- Theoretically, the paper shows that rank decays to one doubly exponentially in depth, similar to softmax attention.
- With experiments, the paper showed that Sinkhorn better preserves rank than softmax on text data (AG's new dataset).

Strengths:

- Characterised the decay rate of Sinkhorn attention theoretically and experimentally, with good ablation studies.
- Paper is well-written

Question:

Is it expected that the normalized spectral norm is small at path depth 0 for SAN (e.g. Figure 3(c), columns 3 & 4)?

**Audience:**

Yes

**Audience Explanation:**

A characterisation of the rank collapse property of an alternative attention mechanism is interesting to the theory community that studies rank collapse. The study may also inform practitioners on the use of Sinkhorn attention.

**Broader Impact Concerns:**

None.

**Claims And Evidence:**

Yes

**Claims Explanation:**

The theory and experiments seem sound and supports the main claims of the paper.

**Requested Changes:**

None.

---

> ### Author Response · Authors · 2026-06-11
>
> We thank the reviewer for their positive assessment of our paper.
>
> Regarding the initial low rank of the attention matrices in Fig. 3 when skip connections are absent: this is consistent with the observation that the rank of hidden states immediately collapses without residual connections, as reported in prior work, for example by Dong et al. (2021), and as also evident in our results in Fig. 4. Note that our main interest is in settings where the rank has not collapsed, for instance due to skip connections, where differences between normalization schemes can still be meaningfully assessed. We have better clarified this point in our minor revision of the paper at page 8, upon the request of Reviewer uwHV.

---

### Review · Reviewer_656r · 2026-06-12

**Summary Of Contributions:**

The work investigates rank collapse in doubly stochastic attention operators in transformers via Sinkhorn iteration, first introduced in Sinkformers (Sander et al. 2022). The paper studies this through SANs, which are networks of only self-attention blocks, deriving theoretical results and norm bounds on Sinkhorn activated attention operators, comparing this to standard softmax activated blocks derived in previous works (Dong et al. 2021). The authors show that Sinkhorn attention has the same asymptotic doubly-exponential collapse as softmax, but shows empirical evidence that Sinkhorn decays slower in early layers.

The paper briefly motivates rank collapse through Dong et. al 2021, mostly through the study of SANs, distinct from Transformer models in practical deployment. If the product of attention matrices converges to rank one, all token representations become identical. The paper confirms that skip connections prevents such a collapse, but does not expand the theoretical derivations to include identity in the skip connections. Importantly, the empirical evaluations focus on models trained as standard transformers, but evaluated as SANs with and without skip connections, as well as standard transformer blocks with FFN.

**Audience:**

No

**Audience Explanation:**

The most significant issue then, is not that the claims do not match the provided evidence, but rather that the results are less clearly linked to practical utility and theoretical understanding of the role of doubly stochastic attention operators vs. row stochastic attention operators in deployment. The motivation for SANs is somewhat under-communicated, and while the authors specify the effect of residual connections, it is somewhat detached from the significance of the study. Additionally, the discrepancy from the theoretical study and empirical settings is not clearly explained, which could potentially lead to misreadings.

Generally, my reading of the paper is missing a general motivation of the study. The paper motivates the study through rank collapse in SANs, but the necessary link to existing models is not provided. The authors state:

> Transformers are the state-of-the-art architecture for large language models [...]

so the motivation here is very implicit, and not fully laid out to the reader. This is in addition to perceived ambiguity on how tight the presented bounds are, as well as the discrepancy from pure SANs in the theoretical study, to residual variants.

In summary, I feel there are several under-communicated approximations in the derivations and results, and that this makes the contributions unclear. For the paper to properly engage with TMLR readers, these need to be clarified explicitly. This is not a point on novelty, but on positioning the paper clearly in the literature, as well as an independent text for the reader. Hence, I believe this needs to be addressed before the paper can be accepted.

**Broader Impact Concerns:**

The paper focuses on theoretical advancements by a specific approximation for doubly stochastic matrices in attention operators. As a result, I see no need for a broader impact statement.

**Claims And Evidence:**

Yes

**Claims Explanation:**

The focus on the paper is SANs, which are distinct from Transformers, and not an architecture in practical use. The authors also carefully point to the fact that skip connections typically help guard against rank collapse, but does not include this setting in the theoretical analysis. It is not entirely clear why SAN + residuals is not the focus, since the authors admit that this is the setting of interest (yet still an approximation).

The theoretical results use a first order approximation of Sinkhorn iteration (projection $Q$, Lem. 1). Hence, the bound presented has an approximation error that is not clearly discussed in the paper. While this may be inherited from previous work, this should still be clarified.
Thm. 2 has the factor raised to $(3^L - 1)/2$. For $L=24$, that is equivalent to `1.4e11` which is very large indeed. Unless the base is extremely close to zero, the bound then seems to be meaningless. Moreover, the authors never evaluate the bound numerically for their experimental settings.

One clear discrepancy for the empirical study in Sec. 4 is that the weights are trained as a transformer with FFNs, but is later evaluated as a pure SAN w. / w.o. residuals at inference.  The attention operators were learned as part of a full transformer architecture. Studying their products in isolation does not necessarily tell the reader what a model trained as a pure SAN would do. The authors do not (as far as I can tell) acknowledge this confounder, and as a result, it weakens the link between empirical results and theory. More on this below.

Despite some issues, I believe the work can be clarified to support the claims made in the paper, with some additional revisions by the authors, hence I lean towards yes on the point of soundness.

**Requested Changes:**

Generally, I would advice the authors to clearly lay out assumptions, approximations, and specify the relevance of SANs to Transformers more clearly. It would also be prudent to explicitly point out why the theoretical results are presented for SANs without residuals. I would also suggest adding an explicit limitations section, which allows the reader to contextualize the study in the broader field of research.

### Questions:

- Q1: Is there a reason why the authors omit a formal analysis of the SAN + Residual connections? This should be clarified and discussed in the paper more explicitly.
- Q2: As far as I could tell, error bounds for the first order Sinkhorn approximation are not included. How does that affect the results? The authors should ideally discuss the effect of the approximation on the result.
- Q3: The $\lambda$ in Prop. 3 seems critical for the argument, which --- as I understand --- depends on how far $Y$ is from $\text{TDS}_{n\times n}$. This seems linked to the learned weights, which is an assumption that could be empirically motivated perhaps? Can the bound be numerically estimated?
- Q4: Can the authors comment on the $(3^L - 1)/2$ bound in Thm.2?

---

> ### Author Response · Authors · 2026-06-24
>
> We thank the reviewer for the valuable and constructive comments. We provide point-by-point responses in the following and upload a revised version of the manuscript with changes highlighted in blue. Following the reviewer's comments, we have added a theorem on the rank decay bound for the case of SAN with skip connections and an explicit Limitations section immediately before the Conclusions.
>
> (1) Formal analysis of SAN with skip connections:
>
> We thank the reviewer for this insightful question. We have now added Theorem 7 in Appendix C, following Dong et al. (2021), focusing on the case with skip connections. As demonstrated in this result, and as expected, the skip connections greatly help in slowing down rank decay.
>
> (2) Error bounds for the first order Sinkhorn approximation:
>
> We thank the reviewer for raising this point. The first-order Sinkhorn approximation in Lemma 1 introduces an error term $o(t)$ involving higher-order softmax derivatives whose denominators scale with powers of $n$ greater than 2, making them negligible. We have added a clarifying remark at the end of Appendix C.
>
> (3) Role of $\lambda$ in Proposition 3:
>
> While $\lambda$ does depend on the distance of the attention matrix from the set of doubly stochastic matrices, the critical quantity for our theoretical result is the entire contraction coefficient $\frac{\lambda\ \beta\ H}{\sqrt{n^{3}d_{qk}}}$, which must be strictly less than 1. Since $0 < \lambda \leq 1$ by construction and $n, d_{qk} \gg 1$, even in the worst case ($\lambda = 1$), the factor $\frac{\lambda}{\sqrt{n^{3}d_{qk}}}$ remains well below one. We expect $\lambda$ to vary empirically depending on the learned weight matrices, but this variation does not affect the qualitative conclusion of our bound.
>
> (4) $(3^L-1)/2$ exponent in Theorem 2:
>
> As noted in the paper at the end of Section 3, because of the factor $\sqrt{n^3}$ in the denominator, if the quantity $\|W_Q W_K^\top\|_2\|W_h\|_2$ remains of order $O(1)$ after training, the coefficient $\frac{\lambda\ \beta\ H}{\sqrt{n^{3}d{qk}}}$ is strictly smaller than 1. For the bound to be meaningful, we only require the base of the exponent to be less than 1. Nevertheless, we agree that the exponent $(3^L - 1)/2$ causes the bound to decay rapidly toward zero as depth increases. However, our focus is on the qualitative behavior captured by the exponent, which matches the same double exponential decay appearing in the analysis of Dong et al. (2021) for the softmax case.
>
> (5) Training as a full Transformer with FFN, while inference as SAN without FFN:
>
> We thank the reviewer for this observation. Our evaluation protocol was chosen to align with the methodology of Dong et al. (2021), enabling direct comparison with their results. We agree that attention weights learned within a full Transformer architecture may behave differently than those learned in a pure SAN setting, and we have now added a Limitations section where we explicitly acknowledge this.